# Large Paraumbilical Vein Shunts Increase the Risk of Overt Hepatic Encephalopathy after Transjugular Intrahepatic Portosystemic Shunt Placement

**DOI:** 10.3390/jcm12010158

**Published:** 2022-12-25

**Authors:** Hao-Huan Tang, Zi-Chen Zhang, Zi-Le Zhao, Bin-Yan Zhong, Chen Fan, Xiao-Li Zhu, Wei-Dong Wang

**Affiliations:** 1Department of Interventional Radiology, The Affiliated Wuxi People’s Hospital of Nanjing Medical University, Wuxi 214023, China; 2Department of Interventional Radiology, The First Affiliated Hospital of Soochow University, Suzhou 215006, China; 3Department of Gastroenterology, The First Affiliated Hospital of Soochow University, Suzhou 215006, China

**Keywords:** transjugular intrahepatic portosystemic shunt, paraumbilical vein, spontaneous portal shunts, hepatic encephalopathy

## Abstract

Background: This study aimed to evaluate whether a large paraumbilical vein (L-PUV) was independently associated with the occurrence of overt hepatic encephalopathy (OHE) after the implantation of a transjugular intrahepatic portosystemic shunt (TIPS). Methods: This bi-center retrospective study included patients with cirrhotic variceal bleeding treated with a TIPS between December 2015 and June 2021. An L-PUV was defined in line with the following criteria: cross-sectional areas > 83 square millimeters, diameter ≥ 8 mm, or greater than half of the diameter of the main portal vein. The primary outcome was the 2-year OHE rate, and secondary outcomes included the 2-year mortality, all-cause rebleeding rate, and shunt dysfunction rate. Results: After 1:2 propensity score matching, a total of 27 patients with an L-PUV and 54 patients without any SPSS (control group) were included. Patients with an L-PUV had significantly higher 2-year OHE rates compared with the control group (51.9% vs. 25.9%, HR = 2.301, 95%CI 1.094–4.839, *p* = 0.028) and similar rates of 2-year mortality (14.8% vs. 11.1%, HR = 1.497, 95%CI 0.422–5.314, *p* = 0.532), as well as variceal rebleeding (11.1% vs. 13.0%, HR = 0.860, 95%CI 0.222–3.327, *p* = 0.827). Liver function parameters were similar in both groups during the follow-up, with a tendency toward higher shunt patency in the L-PUV group (*p* = 0.067). Multivariate analysis indicated that having an L-PUV (HR = 2.127, 95%CI 1.050–4.682, *p* = 0.037) was the only independent risk factor for the incidence of 2-year OHE. Conclusions: Having an L-PUV was associated with an increased risk of OHE after a TIPS. Prophylaxis management should be considered during clinical management.

## 1. Introduction

Aggravated portal hypertension triggers complications, such as esophagogastric variceal bleeding and refractory ascites, in the decompensated stage of cirrhosis. In addition, it drives the development of spontaneous portosystemic shunts (SPSSs) and the formation of collateral circulation, including splenorenal shunts (SRSs), gastrorenal shunts (GRSs), paraumbilical veins (PUVs), and portocaval or mesorenal/caval shunts [1,2]. This compensatory decompression mechanism was recently found to decrease hepatic portal perfusion and increase the risk of hepatic encephalopathy (HE) and death [3,4].

By significantly reducing portal vein pressure, the transjugular intrahepatic portosystemic shunt (TIPS) is an effective therapy for acute variceal bleeding, as well as for the prevention of rebleeding in patients with cirrhosis [5,6]. Post-TIPS HE is one of the major complications after TIPS creation, with a 1-year incidence of 23–54.5% [7,8]. Recent studies [1,4,9] demonstrated that the embolization of large SPSSs may be a therapeutic target to reduce the risk of HE after a TIPS, whereas the definitions of large SPSSs are under debate. The major reported types of SPSSs are mostly SRSs or GRSs, whereas PUVs are less discussed [9,10,11].

In the Baveno VI Cooperation Group cohort [4], large SPSSs were observed in 28% of cirrhotic patients, with SRSs and GRSs accounting for more than half of these (55%), and PUVs approaching one-third (27%). Other rare examples included mesocaval shunts (5%), inferior mesenteric vein-caval shunts (4%), and mesorenal shunts (3%). PUVs are located in the falciform ligament and originate from the left portal vein (LPV), which anastomoses with the anterior subcutaneous vein of the abdominal wall. The typical presentation is the formation of a caput medusae appearance [3]. Unlike other extrahepatic collateral circulation systems, PUVs are rarely a source of bleeding associated with portal hypertension but tend to increase the main portal vein (MPV) blood flow [12]. In cirrhotic patients without TIPS placement, PUVs lead to increases in the portal flow volume, HVPG, and grade of esophageal varices but not do increase the risk of death [13]. However, the impact of PUVs, particularly those meeting large SPSS criteria, on clinical outcomes after TIPS implementation is not yet known. Identifying large PUVs (L-PUVs) may have potential significance for patient management.

Therefore, this retrospective study was carried out to compare the differences in primary clinical outcomes, such as overt hepatic encephalopathy (OHE), death, rebleeding, and shunt dysfunction after TIPS implementation in patients with L-PUVs and those without any SPSSs.

## 2. Materials and Methods

### 2.1. Patients

In total, 386 patients who received a TIPS for acute variceal bleeding or to prevent variceal rebleeding from December 2015 to June 2021 were screened in this study. Two hundred and eighteen patients were included in the analysis (Figure 1). Of these, 78 (35.8%) had received β-blockers combined with endoscopic therapy, while the remaining received an emergency TIPS [14].

Regarding the definition of a large SPSS from previous studies [1,4,9], a PUV is defined as large when it satisfies any of the following criteria: cross-sectional areas > 83 square millimeters, diameter ≥ 8 mm, or greater than half of the diameter of the MPV. The patients were classified into two groups: large PUV (L-PUV) and no SPSS (control group).

The inclusion criteria were as follows: age > 18 years and liver cirrhosis (diagnosed based on clinical presentation, laboratory tests, images, or liver biopsies). The exclusion criteria were as follows: previous HE, the presence of PUV but did not meet large PUV criteria, the presence of other SPSS detected using enhanced CT and portal venography, PUV synchronously embolized intraoperatively, bleeding from ectopic varices, previous treatments with a TIPS or surgical shunt, primary liver cancer or extrahepatic tumors, portal cavernous transformation, or incomplete baseline data or loss to follow up.

### 2.2. Radiological Data Acquisition

Preoperative enhanced abdominal CT scans (Revolution CT, GE Healthcare and SOMATOM Force, Siemens Healthcare) were reviewed by a radiologist (at least 10 years experience in the diagnosis of abdominal diseases, such as liver, biliary, and pancreas) and an interventionalist. The preferred portal phase was observed from a top-down perspective in the axial plane images for the presence of SPSSs and verified in the coronal and sagittal planes. It was recorded only when the PUV diameter exceeded 5 mm. The starting point or the position where the vessel leaves the liver was usually chosen, and the maximum diameter of the PUV was measured in the axis position. Then, the cross-sectional area of the PUV was calculated using the formula πr^2^ [1].

The diameter of the MPV was measured at the junction of the superior mesenteric vein and the splenic vein, and the diameters of the LPV and the right portal vein (RPV) were determined at the junctures of the LPV and RPV, respectively. The presence and grade [15] of the esophageal varices were observed and recorded in the axial plane.

### 2.3. Outcomes and Definitions

The primary outcome was the 2-year OHE rate, and secondary outcomes included the 2-year mortality, all-cause rebleeding rate, and shunt dysfunction rate.

HE was diagnosed and graded according to the West Haven criteria [16]. OHE was defined as grade 2 and above, and severe HE was defined as grade 3 and above. Refractory HE was defined as recurrent (at least 3 episodes of HE in the past 3 months) or persistent symptoms (persistently detectable mental status changes with further episodic deterioration) despite sufficient pharmacological treatment [16,17]. Rebleeding was defined as the occurrence of persistent or new clinical symptoms of bleeding according to the Baveno VI consensus [18]. Shunt dysfunction was defined as a very slow (<40 cm/s) or rapid (>200 cm/s) flow velocity in the stent or a low flow velocity in the portal vein (<20 cm/s) according to ultrasonography or as shunt stenosis > 50% according to the portography procedure [19]. Technical success, clinical success, and complications (major and minor) are referenced in the SIR standards [20].

### 2.4. TIPS Procedure and Follow-Up

The TIPS procedures (Appendix A) in both centers were performed by a team leader with over 10 years of experience in TIPS creation. The angiographic equipment comprised Artis one and Artis zee (Siemens, Munich, Germany). The procedure was performed under sedation and local anesthesia. The 10F guiding sheath was inserted through the jugular vein and the Rups-100 suite (Cook Medical, Bloomington, IN, USA) was introduced into the hepatic vein along the guidewire. After puncturing the portal vein, portal venography clarified the blood flow, location, and classification of the gastroesophageal varices. Embolization of the varices with coils (Cook Medical, Bloomington, IN, USA, or Boston Scientific, Marlborough, MA, USA) and/or *n*-butyl-2-cyanoacrylate (NBCA, BME, Guangzhou, China) was performed. NBCA was mixed with lipiodol in a ratio of 1:2 to 1:3. A 6 or 8 mm-diameter balloon (Boston Scientific, Marlborough, Massachusetts, USA, or Abbott, Chicago, IL, USA) was then selected to dilate the shunt until the incision had completely disappeared. All patients were implanted with an 8 mm e-polytetrafluoroethylene (e-PTFE)-covered stent (Viatorr TIPS endoprosthesis, W. L. Gore & Associates, Inc, Newark, USA). The placement of an additional covered stent (Vanbahn, W. L. Gore & Associates, Inc., Newark, NJ, USA, and Wallstent, Boston Scientific, Marlborough, MA, USA) was evaluated depending on the portal vein and superior mesenteric vein thrombosis. Measurement of the portal pressure gradient (ppg) pre- and post-stenting was an essential step.

Patients with viral hepatitis were treated with antiviral therapy after the TIPS placement. HE prophylaxis was not routinely performed after the TIPS implementation. When symptomatic HE was detected, pharmacological treatments with lactulose (10 mL, TID, PO) combined with l-ornithine l-aspartate granules (3 g, TID, PO) or lactulose (400 mg, BID, PO) were generally used. Patients with a suspected TIPS stenosis/occlusion based on imaging or the recurrence of symptoms of portal hypertension (e.g., ascites, variceal bleeding) were hospitalized for TIPS venography and pressure measurement to confirm whether the shunt was revised.

Follow-up visits included laboratory work and an abdominal ultrasound after 1 and 3 months, as well as ultrasound or enhanced CT after 6 months, and every 6 months thereafter. Evaluations were conducted by at least one physician and one nurse via hospitalization, outpatient service, and telephone calls. The follow-up period was defined as the time interval between the initial TIPS placement to death, liver transplantation, or the end of this study.

### 2.5. Statistical Analysis

Continuous variables are presented as the mean ± standard deviation or median and IQR and were compared with Student’s t-test or the Mann–Whitney U test depending on whether they were normally distributed. Categorical variables are expressed as frequencies (percentages) and were compared using the chi-square test or Kruskal–Wallis H test.

Univariate analysis was performed to identify variables that were potentially related to OHE. Cox regression analysis (forward step-wise likelihood quotient) was performed using the significant predictors in the univariate analysis (*p* < 0.1) to determine the independent predictors of 2-year OHE after a TIPS placement. Cumulative rates of time free of OHE and survival were expressed with Kaplan–Meier curves and compared using the log-rank test. The contribution of each parameter to the risk of developing the outcome was recorded as the HR with a 95% confidence interval (CI). To reduce the baseline differences and the probability of selection bias, we performed (1:2) propensity score matching (PSM) for the L-PUV and control groups. Having had a splenectomy or the presence of gastric varices, platelets, or white blood cells were adjusted with a maximum propensity score distance (caliper) of 0.1.

All tests were two-sided, and significance was established at *p* < 0.05. Statistical analyses were performed using the SPSS 26.0 (IBM Corporation, Somers, NY, USA) or R 4.2.2 (http://www.R-project.org/, accessed on 20 October 2022) software packages.

## 3. Results

### 3.1. Patient Characteristics

The characteristics of the patients at baseline are shown in Appendix A. To accurately describe the practical application of the TIPS, it is reported for the pre-matched cohort. In total, 218 patients had a technical success rate of 100% and a clinical success rate of 95.0% (11 cases (5.0%) of variceal rebleeding were confirmed via endoscopy 2 years after the TIPS). According to the incidence, the main complications were as follows: HE (26/218, 11.9%), hepatic failure (8/218, 3.7%), biliary bleeding (3/218, 1.4%), hepatic artery injury (2/218, 0.9%), pneumonia (2/218, 0.9%), vomiting (2/218, 0.9%), abdominal pain (2/218, 0.9%), fever (1/218, 0.4%), and malignant arhythmia (1/218, 0.4%). Minor complications included HE (43/218, 19.7%), abdominal pain (15/218, 6.9%), vomiting (9/218, 4.1%), fever (5/218, 2.3%), transient pulmonary edema (1/218, 0.4%), and enter site hematoma (1/218, 0.4%), sequentially.

L-PUVs were present in 27 (12.4%) patients with a median diameter of 9.8 mm (IQR: 8.8–14.0), of which 23 (10.2%) patients showed a diameter ≥ 8 mm and 25 (11.1%) patients showed a diameter greater than half the diameter of the MPV. The median of the cross-sectional areas was 75.4 square millimeters (IQR: 55.4–158.4), with 13 (5.8%) patients showing an area > 83 square millimeters (Appendix A).

Overall, the pre-matched L-PUV group had larger diameters of the LPV and a high presence of gastric varices. None of these patients underwent a splenectomy and had lower platelet and white blood cell counts. There were no statistically significant differences in terms of liver function, imaging parameters, esophageal vein grading, PPG, the diameter of the balloon, or follow-up time in either group.

### 3.2. Propensity Score Matching

To reduce the effect of imbalanced baseline information, 27 patients with L-PUVs were matched to 54 patients without any SPSSs after propensity matching for having had a splenectomy and the presence of gastric varices, platelets, and white blood cells.

As described in Table 1, the baseline characteristics of the two groups were comparable after matching when considering the diameter of the LPV. The median follow-up times were 22.0 months (IQR: 12.0–43.0) and 23.0 months (IQR: 14.7–39.0) in the L-PUV and control groups, respectively (*p* = 0.996). One patient developed primary liver cancer during follow-up in the control group.

### 3.3. Overt Hepatic Encephalopathy

Fourteen patients in each of the L-PUV and the control groups (51.9% vs. 25.9%, *p* = 0.022) experienced at least one episode of OHE within 2 years (Table 2). The average numbers of OHE episodes were 0.9 ± 1.2 and 0.7 ± 1.3 in the two groups, respectively (*p* = 0.426). Severe HE (grade III/IV) occurred in three (11.1%) patients in the L-PUV group and seven (13.0%) patients in the control group (*p* = 0.810), of which one (3.7%) and five (9.3%) patients, respectively, developed refractory HE (*p* = 0.342). All OHE patients were hospitalized for treatment. No patient received additional PUV embolization during the follow-up. Except for one patient in the control group who received shunt reduction, the remaining patients were treated with lactulose combined with ornithine methylate pellets or rifaximin after discharge.

As shown in Figure 2A, the 1-year actuarial probability of OHE in the L-PUV group was similar to the control group (33.3% vs. 25.9%, HR = 1.413, 95%CI 0.612–3.266, log-rank test: *p* = 0.406), but the 2-year actuarial probability of OHE was significantly higher (51.9% vs. 25.9%, HR = 2.301, 95%CI 1.094–4.839, log-rank test: *p* = 0.021).

Univariate and multivariate analyses were performed on all matched patients (Table 3). Factors affecting the occurrence of OHE at 2 years included total bilirubin and L-PUV. Multifactorial analysis adjusted for confounders showed L-PUV (HR = 2.600, 95%CI 1.148–5.889, *p* = 0.022) was the only independent influencing factor for 2-year OHE.

### 3.4. Mortality, Rebleeding, and Shunt Dysfunction

Compared with the control group, the L-PUV group showed no significant difference between the 1-year (14.8% vs. 7.4%, HR = 2.155, 95%CI 0.539–8.618, log-rank test: *p* = 0.256) and 2-year (14.8% vs. 11.1%, HR = 1.497, 95%CI 0.422–5.314, log-rank test: *p* = 0.529) actuarial mortality probabilities (Figure 2B). The detailed causes of death are listed in Table 2. There was no statistical difference between the 1-year (7.4% vs. 13.3%, HR = 0.703, 95%CI 0.142–3.483, log-rank test: *p* = 0.664) and 2-year (11.1% vs. 13.3%, HR = 0.860, 95%CI 0.222–3.327, log-rank test: *p* = 0.827) actuarial rebleeding probabilities between the L-PUV and control groups. Reasons for rebleeding are shown in Table 2.

Two (3.7%) and four (5.6%) patients in the control group had shunt dysfunction at 1 and 2 years, respectively; however, no dysfunction occurred in the L-PUV group during the follow-up. Of these four patients, one was treated with balloon dilation combined with thrombolysis, and three were treated with stent implantation. All of them had normalized shunt blood flow after the procedure. Except for these patients, no ascites deterioration was found.

### 3.5. Changes in Liver Function and L-PUV Diameter

The total bilirubin, Child–Pugh score, and MELD score in both groups showed overall elevated and then decreased changes, and albumin improved ineffectively after the procedure. Creatinine was high in the control cohort 1 month after the TIPS. Overall, there was no significant difference in liver function indicators between the two groups at each time point of observation (Figure 3).

At 6 months (9.7 (7.0–13.6), *p* < 0.001), 1 year (8.9 (5.9–12.3), *p* < 0.001), and 2 years (7.8 (3.0–11.1), *p* = 0.024) after TIPS implantation, statistically significant changes in the L-PUV diameter were observed compared with the previous follow-up (Figure 4). There was no new or increased diameter of the PUV in either group by the end of the study. The L-PUV showed a gradual collapse post-TIPS, but only completely disappeared in five (18.5%) patients (Appendix A).

## 4. Discussion

Our study demonstrated that patients with variceal bleeding in unembolized L-PUVs had no increased risk of death, rebleeding, or worsening liver function 2 years after TIPS implementation, but did have a 2.2-fold increased risk of OHE 2 years after the operation. This study highlighted the robustness of L-SPSSs in predicting OHE, further refined the impact of the L-SPSS classification on clinical outcomes after TIPS implementation, and provided a reference for the prophylactic treatment of patients at high risk of OHE.

Yi F. et al. [21] found that SRSs may narrow portal vein diameter (RPV and MPV) and shrink liver volume, impairing hepatic function and survival. In contrast, we found a larger LPV diameter in L-PUV patients, which may have been caused by different hemodynamic characteristics. Unlike SRSs, the PUV allows blood to pass through the MPV rather than leave it, hence the term right-sided shunt [22]. The current recognition of PUVs is still controversial. One idea is that the PUV benefits the maintenance of MPV volume and velocity, preserves MPV diameter, and reduces the risk of portal thrombosis [23]. The other is that the PUV has no protective effect on hepatic perfusion, but deteriorates liver function [12,24]. However, whether from the perspective of increasing portal perfusion or worsening liver function, PUV may increase the risk of OHE after TIPS implementation.

Similar to several previous studies [9,10,25], we confirmed that there was indeed an association between an L-PUV and OHE. This indicated that even if the anatomical drainage of the PUV is different from other collateral circulation, a large shunt can still affect OHE after TIPS. Decreases in L-PUV diameter were mainly caused by the normalization of portal vein pressure after the TIPS. This change may be relatively rapid after TIPS placement. However, the collapse of the L-PUV diameter did not occur as quickly as expected, and it only completely disappeared in a quarter of patients by the end of the study. This moderate change may have resulted in liver function and mortality being comparable in both groups. Interestingly, OHE was gradually induced in the L-PUV group. We hypothesized that this may have been related to covert HE (CHE), as some reports [26] discovered that the deterioration of CHE to OHE may be influenced by this shunt.

Notably, an L-PUV did not aggravate the OHE grading. Furthermore, the symptoms of OHE were manageable through routine medical care and did not require additional PUV embolization. Therefore, we cannot recommend L-PUV embolization based on the present study; in previous studies, the short- and long-term benefits of embolizing large SPSSs are lacking, except in patients with refractory or recurrent HE [27]. Moreover, after receiving large SPSS embolization, a significant proportion of patients still did not completely avoid HE [9]. Moreover, a cost–benefit analysis is required.

After controlling for consistent risk factors [28,29,30], such as age, liver function, history of HE, stent diameter, and diabetes, half of the patients in the L-PUV group still experienced OHE. This suggested that in addition to familiar factors, having an L-PUV was a relatively reliable predictor, which reminded us to consider the necessity of prophylactic treatment for these patients in the absence of embolization of the PUV. Elevated serum bilirubin was a direct indicator of hepatocellular dysfunction and generally tended to increase and then decrease after TIPS implementation. Coinciding with the findings of Fonio P [31] and Lin X [32], we found that bilirubin was associated with post-TIPS OHE but was not an independent risk factor. Our study also assessed under-dilated and completely dilated TIPSs; it may be that under-dilated stent grafts still expand to the nominal diameter with time [33], and thus, we did not find any effect on long-term OHE.

Additionally, L-SPSSs did not cause a decreased survival rate in TIPS patients compared with the cirrhotic cohort without TIPS placement [9,10], and the same results were observed in our study, which could be a survival benefit of TIPS implementation. Although some studies [13,34,35] showed that PUV is associated with alcoholic cirrhosis, esophageal varices grading, and esophageal variceal bleeding, probably limited by case size, in general, we did not find these associations. Moreover, the majority of patients in our study had a punctured LPV. Although choosing a puncture site to reduce the risk of OHE is unreliable [36], it is unclear whether there is a hemodynamic difference between a large shunt and a stent located ipsilateral and heterolateral.

The study had several limitations. First, the sample size of the study was small, which could have led to selection bias. Second, limited by the invasive nature of the assay, we were unable to assess the changes in PPG after the collapse of the L-PUV. Third, the study lacked the measurement of flow rates at different sites in the portal vein, as well as liver perfusion, which needs to be improved in future studies. Fourth, we did not compare our results with other SPSSs. However, this is one of the research directions we are currently exploring. Finally, we only selected patients with variceal bleeding and did not include refractory ascites, and thus, the conclusions drawn need to be interpreted with caution.

## 5. Conclusions

In summary, L-PUVs increased the risk of post-TIPS OHE. Even if the overall symptoms are manageable, the prophylactic treatment of OHE in such patients still needs to be considered.

## Figures and Tables

**Figure 1 jcm-12-00158-f001:**
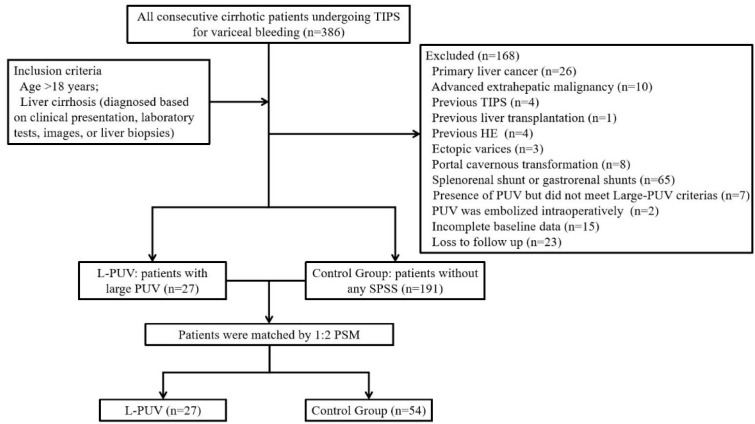
Study flow diagram.

**Figure 2 jcm-12-00158-f002:**
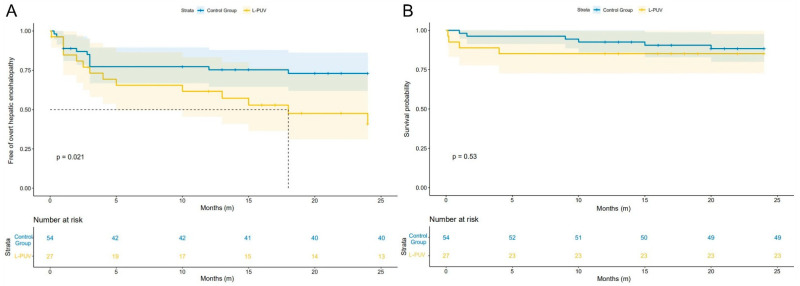
Kaplan–Meier curves of overt hepatic encephalopathy (**A**) and survival (**B**) after TIPS implementation according to the L-PUV and the control groups.

**Figure 3 jcm-12-00158-f003:**
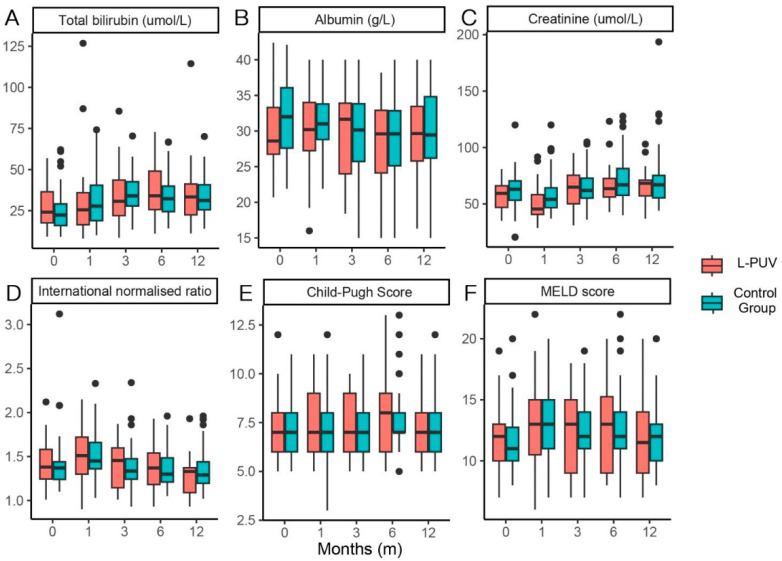
Box plots of total bilirubin (**A**): the *p*-values were 0.344, 0.512, 0.506, 0.624, and 0.913 for pre-TIPS and 1, 3, 6, and 12 months post-TIPS, respectively), albumin (**B**): the *p*-values were 0.110, 0.311, 0.964, 0.729, and 0.817 for pre-TIPS and 1, 3, 6, and 12 months post-TIPS, respectively), creatinine (**C**): the *p*-values were 0.233, 0.032, 0.938, 0.390, and 0.760 for pre-TIPS and 1, 3, 6, and 12 months post-TIPS, respectively), international normalized ratio (**D**): the *p*-values were 0.455, 0.863, 0.599, 0.917, and 0.429 for pre-TIPS and 1, 3, 6, and 12 months post-TIPS, respectively), Child–Pugh score (**E**): the *p*-values were 0.236, 0.645, 0.264, 0.500, and 0.628 for pre-TIPS and 1, 3, 6, and 12 months post-TIPS, respectively), and MELD score (**F**): the *p*-values were 0.279, 0.709, 0.715, 0.573, and 0.701 for pre-TIPS and 1, 3, 6, and 12 months post-TIPS, respectively) in the L-PUV and control groups.

**Figure 4 jcm-12-00158-f004:**
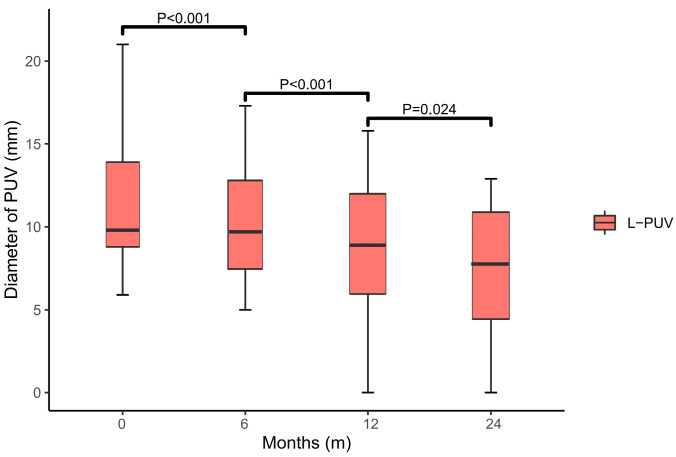
Box plot of the L-PUV diameter variation. There was a significant collapse in the diameter of the L-PUV at 6 (*p* < 0.001), 12 (*p* < 0.001), and 24 (*p* = 0.024) months post-TIPS compared with the previous data.

**Table 1 jcm-12-00158-t001:** Baseline characteristics of patients after PSM.

Parameters	L-PUV (*n* = 27)	Control Group (*n* = 54)	*p*-Value
Sex (male), *n* (%)	15 (55.6%)	28 (51.9%)	0.753
Age (years)	54.0 (42.0–66.0)	55.0 (48.0–66.5)	0.426
Viral hepatitis	15 (55.6%)	30 (55.6%)	1.000
Ascites (present), *n* (%)	13 (48.1%)	29 (53.7%)	0.637
Previous diabetes, *n* (%)	4 (14.8%)	8 (14.8%)	1.000
Spleen diameter (mm)	167.0 (133.9–185.9)	167.5 (150.4–199.5)	0.241
Presence of gastric varices, *n* (%)	5 (18.5%)	7 (13.0%)	0.422
Portal vein thrombosis, *n* (%)	7 (25.9%)	16 (29.6%)	0.726
Total bilirubin (μmol/L)	24.1 (16.0–39.5)	22.4 (15.8–29.6)	0.344
Albumin (g/L)	28.6 (26.7–33.6)	32.0 (27.5–36.1)	0.110
Creatinine (μmol/L)	59.4 (45.9–67.2)	63.0 (53.0–70.1)	0.233
Prothrombin time (S)	16.2 (14.7–18.5)	15.6 (14.4–17.6)	0.370
International normalized ratio	1.38 (1.23–1.60)	1.37 (1.24–1.45)	0.455
Platelet count (×10^9^/L)	59.0 (40.0–73.0)	50.0 (34.0–69.3)	0.431
White blood cell (×10^9^/L)	3.3 (2.0–7.23)	2.6 (1.8–5.8)	0.304
Child–Pugh score (points)	7.0 (6.0–8.0)	7.0 (6.0–8.0)	0.236
Child–Pugh grade, *n* (%)			0.226
A	8 (29.6%)	23 (42.6%)	
B	15 (55.6%)	26 (48.1%)	
C	4 (14.8%)	5 (9.3%)	
MELD score (points)	12.0 (10.0–13.0)	11.0 (9.8–13.0)	0.279
Diameter of LPV	13.2 (10.7–14.8)	11.1 (10.0–12.6)	0.010
Diameter of RPV	11.6 (8.1–13.0)	11.0 (9.3–12.3)	0.970
Diameter of MPV	16.9 (13.8–20.6)	16.4 (14.2–18.4)	0.702
Esophageal variceal grade			0.465
0	2 (7.4%)	2 (3.7%)	
I	1 (3.7%)	0	
II	2 (7.4%)	5 (9.3%)	
III	22 (81.5%)	47 (87.0%)	
Portal vein puncture			0.648
LPV	19 (70.4%)	37 (68.5%)	
RPV	7 (25.9%)	9 (16.7%)	
MPV	1 (3.7%)	8 (14.8%)	
Pre-TIPS PPG (mmHg)	21.0 (18.0–21.0)	21.9 (17.8–25.0)	0.099
Post-TIPS PPG (mmHg)	7.0 (6.0–9.8)	8.0 (6.0–10.0)	0.536
Diameter of balloon catheters, *n* (%)			0.695
6 mm	6 (22.2%)	10 (18.5%)	
8 mm	21 (77.8%)	44 (81.5%)	
Duration of follow-up (months)	22.0 (12.0–43.0)	32.0 (20.0–47.3)	0.111

Note: Data are expressed as median (IQR) or *n* (%). Abbreviations: PUV—paraumbilical vein; MELD—model for end-stage liver disease; LPV—left portal vein; RPV—right portal vein; MPV—main portal vein; TIPS—transjugular intrahepatic portosystemic shunt; PPG—portal pressure gradient.

**Table 2 jcm-12-00158-t002:** Summary of outcome measurements within 2 years.

Outcome	L-PUV (*n* = 27)	Control Group (*n* = 54)	*p*-Value
**OHE, *n* (%)**	14 (51.9%)	14 (25.9%)	0.022
Episodes per patient	0.9 ± 1.2	0.7 ± 1.3	0.426
Frequency of OHE			0.056
One episode	9 (33.3%)	5 (9.3%)	
More than one episode	5 (18.5%)	9 (16.7%)	
Severe HE (grade III/IV)	3 (11.1%)	7 (13.0%)	0.810
Refractory HE	1 (3.7%)	5 (9.3%)	0.342
1-month OHE, *n* (%)	4 (14.8%)	9 (16.7%)	0.830
3-month OHE, *n* (%)	7 (25.9%)	13 (24.1%)	0.856
6-month OHE, *n* (%)	9 (33.3%)	13 (24.1%)	0.382
1-year OHE, *n* (%)	9 (33.3%)	14 (25.9%)	0.489
**Death, *n* (%)**	4 (14.8%)	6 (11.1%)	0.637
Cause of death			
Gastrointestinal bleeding	2 (7.4%)	2 (3.7%)	
Liver failure	1 (3.7%)	2 (3.7%)	
Sepsis/pneumonia	1 (3.7%)	1 (1.9%)	
Unrelated to liver disease	0	1 (1.9%)	
**Rebleeding, *n* (%)**	3 (11.1%)	7 (13.0%)	0.810
Sources of bleeding			
Variceal rebleeding	2 (7.4%)	3 (5.6%)	
Portal hypertensive gastropathy	0	2 (3.7%)	
Unknown	1 (3.7%)	2 (3.7%)	
**Shunt dysfunction, *n* (%)**	0	4 (7.4%)	0.067

Note: Data are expressed as mean ± SD or *n* (%). Abbreviations: OHE—overt hepatic encephalopathy; HE—hepatic encephalopathy; PUV—paraumbilical vein.

**Table 3 jcm-12-00158-t003:** COX regression analysis for 2-year OHE after PSM.

Variables	Univariate Analysis			Multivariate Analysis		
	HR	95%CI	*p*-Value	HR	95%CI	*p*-Value
Total bilirubin	1.029	1.001–1.058	0.043			
L-PUV	2.301	1.094–4.839	0.028	2.217	1.050–4.682	0.037

Abbreviations: OHE—overt hepatic encephalopathy; L-PUV—large paraumbilical vein.

## Data Availability

The dataset used and analyzed in the present study are available from the corresponding author on reasonable request.

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
