# Peer review of "Large Paraumbilical Vein Shunts Increase the Risk of Overt Hepatic Encephalopathy after Transjugular Intrahepatic Portosystemic Shunt Placement"

_jcm, 2022, doi:10.3390/jcm12010158_

Round 1
Reviewer 1 Report
I would like to suggest small corrections in the definitions in the introduction and comments are directly inserted in the attached .pdf.

Author Response
Response to Reviewers for jcm-2016938 “Large Paraumbilical Vein Shunts Increase the Risk of Overt Hepatic Encephalopathy after Transjugular Intrahepatic Portosystemic Shunt Placement” by Hao-Huan Tang et al.
(original reviewers’ comments in black, our reponse in red)
Reviewer #1
- “Portosystemic”
Reply: Many thanks, This has been modified accordingly.
- “paraumbilical patent vein or recanalized paraumbilical vein”
Reply: Many thanks for highlighting this. We suggest that neither of these descriptions may be accurate.
Up to now, PUV has been reported in many literatures. There are indeed two descriptions of it, as stated by the reviewer, i.e., “patented” or “recanalized”? With the intensive study of anatomy, we consider that the controversy between these two descriptions is actually for the umbilical vein rather than the paraumbilical vein.
In the fetal circulation, the umbilical vein extends along the margin of the falciform ligament toward the liver, passes into the umbilical fissure, and enters the left branch of the portal vein. Within a few days of normal breathing after birth, the umbilical vein and venous conduit collapsed and were eliminated, becoming the round and venous ligaments, respectively.
Previously, it was believed that portal hypertension resulted in "recanalization" of the occluded umbilical vein, forming a shunt in the portal system to reduce pressure. However, M Lafortune et al.(Am J Roentgenol. 1985;144(3):549-553.) in 1985 confirmed by detailed angiographic and histopathological studies that recanalization of the occluded umbilical vein did not actually occur. The large venous structures moving from the liver to the umbilicus were actually enlarged paramedian veins, 2 or 3 of which were normally present in or adjacent to the falciform ligament. Similarly, the patency of another portal shunt "venous conduit" was rarely observed in adults with advanced portal hypertension, because there were no additional small veins adjacent to this structure.
Of note in this study was that the normal falciform ligament contains 1 to 3 tiny collapsed paraumbilical veins. It would also be understood that the structure of the PUV may be always present. Only when the portal vein pressure increases to a certain degree, an increased number and diameter of the paraumbilical veins occurred. Therefore, we suggest that the description of “patented” or “recanalized” is not accurate. Thus, in the text we describe it directly as paraumbilical vein.
- shunt/patent vein is equivalent to varices???
Reply: Many thanks. The PUV belongs to the collateral pathway of portal hypertension, not the varices. We have modified the text to read (line 66) : "PUV is rarely a source of bleeding associated with portal hypertension".
- but a reduce an effective portal liver perfusion (portal blood flow--paraumbilical blood flow)
Reply: Many thanks. The presence of PUV would theoretically decrease hepatic portal perfusion. Unfortunately, this examination of portal perfusion has not been routinely performed and no relevant studies were retrieved. However, our group is trying to evaluate the changes in hepatic portal perfusion before and after TIPS placement using Revolution CT ( GE Healthcare). Nevertheless, we have not yet found a way to quantify the effect of PUV and hope you can make some valuable suggestions.
- correct conclusion of the quoted manuscript should be revised: these authors showed an increase of: flow volume in the portal vein (assessed by Doppler), HVPG and grade of esophageal varices
Reply: Apologies that this was not adequately clear. We have modified the text to read (lines 68-69): “PUV caused an increase in portal flow volume, HVPG and grade of esophageal varices , but did not raise the risk of death”.
- Shunt dysfunction was defined as a very slow (< 40 cm/sec) or rapid (> 200 cm/sec) flow velocity in the stent or a low flow velocity in the portal vein (< 20 cm/sec) according to ultrasonography, or a shunt stenosis > 50% in the portography.
Reply: We have to apologize, but we didn't see the suggestion you marked here.
- Which are the clinical evidence for esophageal varices embolization?
Reply: Many thanks for raising this important point. Previous studies have suggested that adjunctive gastroesophageal variceal embolization during TIPS procedures might be beneficial in the prevention of variceal rebleeding (J Gastroenterol Hepatol. 2014;29(4):688-696. ). However, it has to be acknowledged that the study has certain limitations, such as inconsistency in the type of stent, embolic agent and type of varices.
Interestingly, a recent RCT from Chinese colleagues did not find a combination variceal vein embolization benefit, but the study using non-Viatorr stents. Additionally, the risk-to-benefit ratio of embolization in certain subgroups, such as patients with inadequate PPG decline after TIPS or those with gastric fundic varices, is worth considering separately (Lancet Gastroenterol Hepatol. 2022;7(8):736-746. ).
Reviewer 2 Report
Dear authors, I have some questions and comments:
Do you measure PUV previously to TIPS?
Do you correlate de PUV diameter with the PPG after TIPS?
Any of these patients received b-blockers ?
Any patient received lactulose or rifaximin prophylactic after TIPS.? It should be reported in table characteristics.
Author Response
Response to Reviewers for jcm-2016938 “Large Paraumbilical Vein Shunts Increase the Risk of Overt Hepatic Encephalopathy after Transjugular Intrahepatic Portosystemic Shunt Placement” by Hao-Huan Tang et al.
(original reviewers’ comments in black, our reponse in red)
Reviewer #2
- Do you measure PUV previously to TIPS?
Reply: Many thanks. To maintain consistency with postoperative CT follow-up measurements, PUV diameters were measured on all cross-sectional CT images in this study.
- Do you correlate de PUV diameter with the PPG after TIPS?
Reply: Very many thanks for these important points. We did not find a correlation between PPG and PUV after TIPS (Spearman correlation coefficient of -0.08, correlation size of 0.08, P=0.232).
- Any of these patients received b-blockers ?
Reply: Many thanks for highlighting this. We have modified the text to read (lines 81-83): “Of these, 81 (36%) received β-blockers combined with endoscopic therapy before TIPS, while the remaining patients received emergent TIPS directly”.
Although both medical centers have well-established emergency access for EGVB, emergent endoscopy is not yet routinely performed, so only 36%patients in the study received β-blockers combined with endoscopic intervention, while the rest received emergent TIPS (J Vasc Interv Radiol. 2022;S1051-0443(22)01335-5.). None of the patients received β-blocker therapy postoperatively.
- Any patient received lactulose or rifaximin prophylactic after TIPS.? It should be reported in table characteristics.
Reply: Many thanks for highlighting that was not adequately clear. Post-TIPS HE prophylaxis was not routinely performed in this study. This is because prophylactic treatment of HE with lactulose or rifaximin was still controversial until the publication of the study by Bureau C et al. (Ann Intern Med. 2021;174(5):633-640.). We have modified the text to read (lines 156-159) : HE prophylaxis was not routinely performed after TIPS. When symptomatic HE was detected, pharmacological treatments with Lactulose (10 ml, TID, PO) combined with l-ornithine l-aspartate grannules (3 g, TID, PO) or lactulose (400 mg, BID, PO) were generally used.
Reviewer 3 Report
The authors mainly investigated a relationship of large paraumbilical vein shunt (PUV) positive and incidence rate of overt hepatic encephalopathy (OHE) after TIPS in follow up period. Large PUV positive group (n=27) demonstrated higher incidence rate of OHE than that of large PUV negative group (n=54). The mortality, rebleeding, shunt disfunction rates after TIPS were comparable in large PUV positive and negative groups.
Spontaneous portosystemic shunt (SPSS) has been already known as risks of OHE and mortality after TIPS (some papers disagree on mortality). SPSS includes splenorenal, portocaval, mesorenal/caval, and gastro-paraesophageal shunts or PUV. SPSS may collapse after TIPS following normalization of portal vein pressure but the size of SPSS remains unchanged after TIPS in some cases. Moreover, Larger SPSS size is more likely to cause OHE after TIPS. PUV is usually one type of SPSSs but can uniquely increase main portal vein flow positively on US in cirrhosis cases. With an increase in the severity of cirrhosis the incidence of PUV patency rises. The author applied it to TIPS cases in this study.
However, the clinical aim of focusing only large PUV is unclear. This study also does not compare TIPS cases with large PUV to TIPS cases with other SPSSs directly. In addition, in TIPS cases with any types of SPSS, OHE is a complication that needs attention after all. Therefore, the conclusion of this study has relatively weak clinal impact with low novelty. It is also problematic that the detailed TIPS procedures are not described at all. It is difficult to understand the author’s intension in main document. It is necessary to distinguish between quotations from other papers and the content of this research results in the discussion section.
The size of this study is adequate.
The references are Ok.
Abstract
Total revision is needed at the abstract section. More introduction about OHE, SPSSs, and PUV should be mentioned at the background section. Methods and conclusion sections should be also modified.
Introduction
P51. Incidence rates of PUV and other SPSSs in cirrhosis cases should be added.
P55. Why did the authors focus the relation of only large PUV and OHE?
The relation of OHE after TIPS and other SPSSs were not assessed in this study. Clear reason should be mentioned.
Material and Methods
P68. Endoscopic treatment was not attempted before TIPS?
P68. What was TIPS indication criterion? MELD score is used generally. This issue is very important.
Why did TIPSs were performed in only bleeding cases?
P69. What did the authors use in order to evaluate PUV? Ex. Portal venography of TIPS, CT or US
P75. What did the authors use in order to evaluate other SPSSs? WL-PUV group (n=198) didn’t accompany with other SPSSs truly?
Figure 1-4: good.
P82. The authors should state the type of the CT scan machine and details of the CT protocol. The data from two centers was used in this study (in the abstract section, the authors mentioned). It is especially important.
P82. What was the experience of the radiologists? It should be mentioned.
P107. Please explain details of treatments for suspected OHE, rebleeding, and shunt disfunction after TIPS.
P116. “Prophylactic drug therapy” This expression is not proper because OHE treatment was not started until clinical manifestation of OHE appeared. In addition, please also include the specific volume dosage of the drug.
P110-122
Description of TIPS procedure method (ex. TIPS device, sheath size, balloons, and TIPS procedure figures) and complications related to TIPS are highly deficient. It should be improved for the purpose of maintaining its reproducibility. The authors preferred to puncture to LPV in TIPS procedure (in discussion section, the authors mentioned), Please explain it.
Technical success and clinical success should be also defined. In addition, did the authors use anticoagulant after TIPS?
P114. Why did only patients with both HBV and HCV receive antiviral therapy after TIPS placement?
Results
Technical success, clinical success, and complications related to TIPS should be mentioned in the result section. This section is a few confusing.
The parameters in table1 and table 2 should be also mentioned at the materials and methods section.
Please show the parameters that had significant differences and those that did not at the result section.
P145. PUV in 34 patients was corresponded to PUV (>5mm)?
P153-154 “the L-PUV group had larger diameters of the LPV and a high presence of gastric varices”
These are not statistically significant differences.
Table1. Child-pugh score raged from 6 to 8. However, Child-pugh grade ranged A to C corresponding to score (5~15). Please explain it.
Table1. Please explain the indication of use of 6-mm or 8-mm balloon after stent placement at the materials and methods section.
Table1. Did you treat portal vein thrombosis?
Table1. Use of proton pump inhibitor (PPI) is regarded as risk of OHE after TIPS too.
P183. Why one patient of WL-PUV group received shunt reduction?
P202. Univariate and multivariate analyses demonstrated significant differences of L-PUV and total bilirubin on OHE incidence rate. There is lack of discussion about bilirubin. Bilirubin didn’t have cutoff value?
Discussion
L-PUV diameter gradually decreased in Figure 4. But, L-PUV diameter decrease was caused mainly by normalization of portal vein pressure after TIPS. This change could be relatively rapid after TIPS. Please explain it.
How many did patients undergo RPV or LPV puncture in TIPS procedure in all case (N=225)?
This factor affected shunt disfunction and other clinical outcomes?
In addition, L-PUV diameter gradually decreased or collapsed, Therefore, maintenance of hepatic function and comparable mortality rate of in TIPS with L-PUV group may be natural in follow up period. In contrast, OHE in TIPS with L-PUV group was induced gradually. Please explain it.
P261. Introduce of large SPSSs classification on clinical factors (mortality, hepatic insufficiency, OHE) was not attempted in this article. Only PUV of SPSSs was mainly investigated.
P288. Why L-PUV embolization after TIPS was not needed in this study and Why OHE could be controlled with conservative therapy alone? Is it caused by hemodynamic specificity of L-PUV?
Conclusion
“Even if the overall symptoms are manageable, prophylactic treatment of OHE in such patients still needs to be considered”
I could not understand this sentence. OHE treatment was not started until clinical manifestation of OHE appeared in this study.
OHEs in this study were controlled using conservative therapy alone without LPUV embolization.
Therefore, even In TIPS cases with LPUV, OHE control was relatively easy with hepatic function being maintained and mortality rates being comparable to TIPS cases without LPUV. Post-interventional care after TIPS cases with LPUV might be easier than TIPS case with other SPSSs.
Does my above speculation match the author's assertion?
Author Response
Response to Reviewers for jcm-2016938 “Large Paraumbilical Vein Shunts Increase the Risk of Overt Hepatic Encephalopathy after Transjugular Intrahepatic Portosystemic Shunt Placement” by Hao-Huan Tang et al.
(original reviewers’ comments in black, our reponse in red)
Reviewer #3
The authors mainly investigated a relationship of large paraumbilical vein shunt (PUV) positive and incidence rate of overt hepatic encephalopathy (OHE) after TIPS in follow up period. Large PUV positive group (n=27) demonstrated higher incidence rate of OHE than that of large PUV negative group (n=54). The mortality, rebleeding, shunt disfunction rates after TIPS were comparable in large PUV positive and negative groups.
Spontaneous portosystemic shunt (SPSS) has been already known as risks of OHE and mortality after TIPS (some papers disagree on mortality). SPSS includes splenorenal, portocaval, mesorenal/caval, and gastro-paraesophageal shunts or PUV. SPSS may collapse after TIPS following normalization of portal vein pressure but the size of SPSS remains unchanged after TIPS in some cases. Moreover, Larger SPSS size is more likely to cause OHE after TIPS. PUV is usually one type of SPSSs but can uniquely increase main portal vein flow positively on US in cirrhosis cases. With an increase in the severity of cirrhosis the incidence of PUV patency rises. The author applied it to TIPS cases in this study.
However, the clinical aim of focusing only large PUV is unclear. This study also does not compare TIPS cases with large PUV to TIPS cases with other SPSSs directly. In addition, in TIPS cases with any types of SPSS, OHE is a complication that needs attention after all. Therefore, the conclusion of this study has relatively weak clinal impact with low novelty. It is also problematic that the detailed TIPS procedures are not described at all. It is difficult to understand the author’s intension in main document. It is necessary to distinguish between quotations from other papers and the content of this research results in the discussion section.
The size of this study is adequate.
The references are Ok.
Abstract
- Total revision is needed at the abstract section. More introduction about OHE, SPSSs, and PUV should be mentioned at the background section. Methods and conclusion sections should be also modified.
Reply: Very many thanks for these important points. We have made detailed revisions to the abstract section and looking forward to your valuable comments.
Introduction
- Incidence rates of PUV and other SPSSs in cirrhosis cases should be added.
Reply: Many thanks for these important points. We have added the following to the text (lines 59-62): “In the Baveno VI Cooperation Group cohort [4], large SPSS were observed in 28% of cirrhotic patients, of which SRS and GRS accounted for more than half (55%), PUV approaching one third (27%), and rare Mesocaval (5%), inferior mesenteric vein-caval (4%),and Mesorenal (3%).”.
- Why did the authors focus the relation of only large PUV and OHE? The relation of OHE after TIPS and other SPSSs were not assessed in this study. Clear reason should be mentioned.
Reply: Many thanks for highlighting this. Splenorenal and Gastrorenal are the most common SPSSs and have been reported many times, while PUV has been less studied. Focusing only on the relationship between PUV and OHE, one is to reduce the interference of other SPSSs and increase the clinical impacts of the findings as much as possible in the case of small sample size; the other is to improve the understanding of PUV and increase the novelty. Finally, the comparison with other SPSSs is one of the directions that the group is researching and will be further improved by future.
Material and Methods
- Endoscopic treatment was not attempted before TIPS?
Reply: Apologies for this oversight. We have modified the text to read (lines 81-83): “Of these, 81 (36%) received β-blockers combined with endoscopic therapy before TIPS, while the remaining patients received emergent TIPS.”.
Although both medical centers have well-established emergency access for EGVB, emergent endoscopy is not yet routinely performed, so only 36%patients in the study received β-blockers combined with endoscopic intervention, while the rest received emergent TIPS (J Vasc Interv Radiol. 2022;S1051-0443(22)01335-5.). This is our latest published study in which the safety and efficacy of emergent TIPS have been confirmed.
- What was TIPS indication criterion? MELD score is used generally. This issue is very important. Why did TIPSs were performed in only bleeding cases?
Reply: Many thanks. The indications for TIPS were treatment of acute variceal bleeding and prevention of rebleeding. While the MELD score was important, it might have been better to use a multidisciplinary approach rather than absolute MELD cutoffs to assess TIPS candidacy (Clin Gastroenterol Hepatol. 2022;20(8):1636-1662.e36.).
Patients selection depends on the characteristics of medical centers. Both hospitals included in the study established emergency green channels for gastrointestinal bleeding, and patients were predominantly EGVB. In contrast, patients with ascites were mostly attended at the local hospital specializing in liver disease. On the other hand, only EGVB cases were included in order to reduce patient heterogeneity. We suggest that further studies could include various types of patients for deeper exploration.
- What did the authors use in order to evaluate PUV? Ex. Portal venography of TIPS, CT or US
Reply: Many thanks. To maintain consistency with postoperative CT follow-up measurements, PUV diameters were measured on all cross-sectional CT images in this study.
- What did the authors use in order to evaluate other SPSSs? WL-PUV group (n=198) didn’t accompany with other SPSSs truly?
Reply: Very many thanks for these important points. We used enhanced CT combined with portal venography to evaluate SPSSs. Any SPSSs other than PUV were excluded, so no other were contained in the WL-PUV group. We have added a detail to the exclusion criteria (lines 90-91): “presence of other SPSS detected by enhanced CT and portal venography.”.
Figure 1-4: good.
- The authors should state the type of the CT scan machine and details of the CT protocol. The data from two centers was used in this study (in the abstract section, the authors mentioned). It is especially important.
Reply: Many thanks. The CT scanners are all Dual-energy CT (Revolution CT, GE Healthcare and SOMATOM Force, Siemens Healthcare). CT protocol details are set according to the respective accompanying manuals.
For space limitations, we have modified the text to read (lines 98-99): “Revolution CT, GE Healthcare and SOMATOM Force, Siemens Healthcare.”.
- P82. What was the experience of the radiologists? It should be mentioned.
Reply: Many thanks. We have modified the text to read (lines 99-100): “Preoperative enhanced abdominal CT was reviewed by a radiologist (at least 10 years experience in the diagnosis of abdominal diseases such as liver, biliary and pancreas) and an interventionalist.”.
- Please explain details of treatments for suspected OHE, rebleeding, and shunt disfunction after TIPS.
Reply: Many thanks for highlighting this. We have added the following to the text (lines 156-162): “HE prophylaxis was not routinely performed after TIPS. When symptomatic HE was detected, pharmacological treatments with Lactulose (10 ml, TID, PO) combined with l-ornithine l-aspartate grannules (3 g, TID, PO) or lactulose (400 mg, BID, PO) were generally used. Patients with suspected TIPS stenosis/occlusion based on imaging or recurrence of symptoms of portal hypertension (e.g., ascites, variceal bleeding) were hospitalized for TIPS venography and pressure measurement to confirm whether the shunt was revised.”.
- “Prophylactic drug therapy” This expression is not proper because OHE treatment was not started until clinical manifestation of OHE appeared. In addition, please also include the specific volume dosage of the drug.
Reply: Apologies that this was not adequately clear. We have modified the text to read (lines 156-159): “HE prophylaxis was not routinely performed after TIPS. When symptomatic HE was detected, pharmacological treatments with Lactulose (10 ml, TID, PO) combined with l-ornithine l-aspartate grannules (3 g, TID, PO) or lactulose (400 mg, BID, PO) were generally used.”.
- P110-122. Description of TIPS procedure method (ex. TIPS device, sheath size, balloons, and TIPS procedure figures) and complications related to TIPS are highly deficient. It should be improved for the purpose of maintaining its reproducibility.
Reply: Thank you very much for meticulous advice. We have added the following to the text (lines 130-147): “The TIPS procedures in both centers were each performed by a team leader with over 10 years of experience in TIPS creation. The angiographic equipment were Artis one and Artis zee (Siemens, Germany). The procedure was performed under sedation and local anesthesia. The 10F guiding sheath was inserted through the jugular vein, and the Rups-100 suite (Cook Medical, Bloomington, Indiana, USA) was introduced into the hepatic vein along the guidewire. After puncturing the portal vein, portal venography clarifies the blood flow, location and classification of gastroesophageal varices. Embolization of varices with coils (Cook Medical, Bloomington, Indiana, USA or Boston Scientific, Marlborough, Massachusetts, USA) and/or n-butyl-2-cyanoacrylate (NBCA, BME, Guangzhou, China). NBCA was mixed with lipiodol in a ratio of 1:2 to 1:3. A 6- or 8-mm diameter balloon (Boston Scientific, Marlborough, Massachusetts, USA or Abbott, Chicago, Illinois, USA) was then selected to dilate the shunt until the incision was completely disappeared. All patients were implanted with an 8 mm e-polytetrafluoroethylene (e-PTFE)-covered stent (Viatorr TIPS endoprosthesis, W. L. Gore & Associates, Inc, Newark, USA). xPlacement of additional covered stent (Vanbahn, W. L. Gore & Associates, Inc., Newark, USA and Wallstent) was evaluated depending on portal vein and superior mesenteric vein thrombosis. Measurement of the portal pressure gradient pre- and post-stenting was an essential step.”.
In addition, we added a detailed TIPS procedure figures (Supplementary figure 1).
- The authors preferred to puncture to LPV in TIPS procedure (in discussion section, the authors mentioned), Please explain it.
Reply: Many thanks. Portal vein puncture was mainly based on the anatomical location of the vessel and reference to current clinical studies. Because there is a debate on the prognostic impact of puncturing the left or the right branch.
- Technical success and clinical success should be also defined.
Reply: Apologies that this was not adequately clear. Technical success describes the successful creation of a shunt between the hepatic vein and intrahepatic branch of the portal vein. Clinical success was defined as control of variceal bleeding symptoms and was determined at the end of each patient follow-up, with rebleeding requiring confirmation by endoscopy. (J Vasc Interv Radiol. 2016;27(1):1-7.)
For space limitations, we have modified the text to read (lines 125-126): “Technical success, clinical success, and complications (major and minor) were referenced to the SIR Standards.”.
- In addition, did the authors use anticoagulant after TIPS?
Reply: Anticoagulation was generally used in the following two situations: first, when there was a stent stenosis or occlusion that needed to be repaired; second, when the PVT was long and affected the blood flow in the portal system, requiring additional stent placement. All were anticoagulated with low molecular weight heparin (LMWH) during hospitalization and treated with oral warfarin (before September 2017) or rivaroxaban (after September 2017) after discharge for a typical duration of 3-6 months, depending on whether the patient had anticoagulation complications and blood flow recovery.
- Why did only patients with both HBV and HCV receive antiviral therapy after TIPS placement?
Reply: Apologies for this oversight. We have modified the text to read (lines 154-155): “Patients with viral hepatitis were treated with antiviral therapy after TIPS placement.”.
Results
- Technical success, clinical success, and complications related to TIPS should be mentioned in the result section. This section is a few confusing.
Reply: We have added the following to the text (lines 190-200): “To accurately described the practical application of TIPS, it was reported in the pre-matched cohort. 225 patients had a technical success rate of 100% and a clinical success rate of 95.1% (11 cases (4.9%) of variceal rebleeding confirmed by endoscopy 2 years after TIPS). According to the incidence, the main complications were: HE (26/225, 11.6%), hepatic failure (8/225, 3.6%), biliary bleeding (3/225, 1.3%), hepatic artery injury (2/225, 0.9%), pneumonia (2/225, 0.9%), vomiting (2/ 225, 0.9%), abdominal pain (2/225, 0.9%), fever (1/225, 0.4%), and malignant arhythmia (1/225, 0.4%). Minor complications sequentially were HE (44/225, 19.6%), abdominal pain (15/225, 6.7%), vomiting (9/225, 4.0%), fever (5/225, 2.2%), transient pulmonary edema (1/ 225, 0.4%), and enter site hematoma (1/225, 0.4%).”.
- The parameters in table1 and table 2 should be also mentioned at the materials and methods section.
Please show the parameters that had significant differences and those that did not at the result section.
Reply: Many thanks, We have added the following to the text (lines 210-212): “There were no statistically significant differences in liver function and imaging parameters, esophageal vein grading, PPG, diameter of the balloon, or follow-up time in both groups.”
- PUV in 34 patients was corresponded to PUV (>5mm)?
Reply: The PUV was >5 mm in all 34 patients, and the characteristics of the PUV were described in Table S2.
- P153-154 “the L-PUV group had larger diameters of the LPV and a high presence of gastric varices”
These are not statistically significant differences.
Reply: Apologies that this was not adequately clear. We have modified the text to read (line 208): “Overall, the pre-matched L-PUV group had larger diameters of the LPV and a high presence of gastric varices.”.
- Child-pugh score raged from 6 to 8. However, Child-pugh grade ranged A to C corresponding to score (5~15). Please explain it.
Reply: Sorry for not describing this clearly. Child-pugh scores in Table 1 were expressed as median (IQR) rather than range.
- Please explain the indication of use of 6-mm or 8-mm balloon after stent placement at the materials and methods section.
Reply: Many thanks for highlighting this. The use of 6-mm or 8-mm balloon depends more on the experience of the operator. In our communication with Chinese colleagues (Clin Transl Gastroenterol. 2021;12(6):e00376.), we found that the impact of adequate or inadequate stent expansion was more significant in early OHE, because this was the period of high incidence of OHE and the stent was in the process of passive expansion. However, long-term follow-up seems to be relatively difficult to see clinical benefit. Therefore, in practice, there have not been clear criteria or indications for the use of small diameter balloons.
- Did you treat portal vein thrombosis?
Reply: There were some patients in the study treated for PVT. Since TIPS itself facilitates PVT recanalization, our treatment strategy was to not anticoagulate if portal venography flow was satisfactory after stenting. However, if the PVT was extensive and additional stent placement was required, post-TIPS anticoagulation would be combined. Furthermore, for patients with shunt failure due to portal vein or in-stent thrombosis, we combine with stent implantation, balloon dilation or thrombolysis on the basis of anticoagulation.
- Use of proton pump inhibitor (PPI) is regarded as risk of OHE after TIPS too.
Reply: Many thanks for highlighting this. Use of PPI increased the risk of OHE after TIPS, as described by Lewis et al. (J Vasc Interv Radiol. 2019;30(2):163-169.) and Dai R et al (Clin Imaging. 2021;77:187-192.). However, over 60% of patients in these two studies underwent TIPS for ascites, so extending the findings to patients with EGVB requires careful consideration. In particular, Chinese patients with cirrhosis have a predominance of viral hepatitis. In addition, long-term oral PPIs after TIPS was not routinely performed at the two medical centers enrolled.
- Why one patient of WL-PUV group received shunt reduction?
Reply: This patient developed refractory OHE during follow-up, which severely affected the quality of life. After multidisciplinary discussions and incorporating the patient's desires, he finally underwent shunt reduction procedure.
- P202. Univariate and multivariate analyses demonstrated significant differences of L-PUV and total bilirubin on OHE incidence rate. There is lack of discussion about bilirubin. Bilirubin didn’t have cutoff value?
Reply: Many thanks for reminding us of the missing point. We have added the following to the text (lines 363-368): “Elevated serum bilirubin was a direct indicator of hepatocellular dysfunction and generally tended to increase and then decrease after TIPS. Coinciding with the findings of Fonio P [29] and Lin X [30], we found that bilirubin was significantly associated with OHE after TIPS. More than this, the ALBI grading established based on bilirubin and albumin also possesses good performance in predicting HE after TIPS.”.
Bilirubin was analyzed as a continuous variable in the study and its cut-off value has not been further explored.
Discussion
- L-PUV diameter gradually decreased in Figure 4. But, L-PUV diameter decrease was caused mainly by normalization of portal vein pressure after TIPS. This change could be relatively rapid after TIPS. Please explain it.
Reply: Vary many thanks for raising this important point. A recent study (Hepatol Int. 2022;10.1007/s12072-022-10440-6. ) found that delayed PPG after 2-4 days were higher than immediate PPG (12.8 ± 4.2 vs 9.2 ± 2.8 mmHg, p<0.001), with a mean increase of 3.6 mmHg. This may be a result of changes that occur after hemodynamic readaptation of the patient. However, as the invasive nature of the measurement, changes in PPG at long-term follow-up were still not known, which was a limitation of this study. Future researches may be facilitated to explain this change if non-invasive tests with higher accuracy are introduced.
To comprehensively understanding the changes in PUV diameter, we added the following to the discussion (lines 334-337):” L-PUV diameter decrease was caused mainly by normalization of portal vein pressure after TIPS. This change may be relatively rapid after TIPS. However, the collapse of the L-PUV diameter did not occurred as quickly as expected, and only a quarter of patients completely disappeared by the end of the study.”.
- How many did patients undergo RPV or LPV puncture in TIPS procedure in all case (N=225)?
This factor affected shunt disfunction and other clinical outcomes?
Reply: Many thanks for highlighting this. The proportion of portal vein puncture sites was, in order, 145 (64.4%) for the left branch, 43 (19.1%) for the right branch, and 37 (16.4%) for the main trunk. Puncture site selection, in both pre- and post-matched COX analyses, had no effect on OHE. However, other outcomes, such as death, rebleeding, and shunt failure, could not be analyzed statistically rigorously because of their low incidence.
- In addition, L-PUV diameter gradually decreased or collapsed, Therefore, maintenance of hepatic function and comparable mortality rate of in TIPS with L-PUV group may be natural in follow up period. In contrast, OHE in TIPS with L-PUV group was induced gradually. Please explain it.
Reply: Vary many thanks for raising this important point. We have added the following to the text (lines 337-341): ” This moderate change may have maintained liver function and mortality comparable in both groups. Interestingly, OHE was gradually induced in the L-PUV group. We hypothesized that this may be related to covert HE (CHE). Since some reports [24] discovered that the deterioration of CHE to OHE may be influenced by this shunt.”.
- Introduce of large SPSSs classification on clinical factors (mortality, hepatic insufficiency, OHE) was not attempted in this article. Only PUV of SPSSs was mainly investigated.
Reply: We very much agree and have to face the shortcomings of the study. Focusing only on PUV is indeed a limitation of this paper, but the group is currently working in conjunction with other centers to attempt investigations comparing the clinical outcomes of different SPSSs, and will report the latest results in time for follow-up.
For the interpretation of limitations, We have added the following to the text (lines 387-388): “Fourth, it lacked comparison with other SPSS. However, this is one of the directions we are currently exploring.”.
- Why L-PUV embolization after TIPS was not needed in this study and Why OHE could be controlled with conservative therapy alone? Is it caused by hemodynamic specificity of L-PUV?
Reply: Very many thanks for these important points. Firstly, in the early days, we suffered from a lack of awareness of PUVs, focusing more on SRS/GRS, which were more common and the target vessels for BRTO management. With the intensive study of SPSSs, we realized that the relatively low incidence of PUVs seemed to escape our surveillance. This was because most PUVs did not visualize or were very faintly apparent on portal venograms after stent placement. However, we found that many PUVs do persist during follow-up.
Since then, we retrospectively analyzed more than 700 patients, including populations with and without TIPS. Some additional work on TIPS combined with PUV embolization has been done, but the sample size was small and insufficient to support a contrastive research. None of the patients in this study had embolized PUV, in order to more critically observe, without intervention, the impact of L-PUV on clinical outcomes.
Pharmacological therapy can control the symptoms of OHE in patients with L-PUV, as concluded by observation. In fact, our comprehension of the hemodynamic characteristics of the PUV remains at a theoretical stage. However, we are monitoring the changes in diameter, flow rate and blood flow in the main portal vein, the right and left branches of the portal vein, the splenic vein and the PUV in some patients, which we expect to further interpretation through objective indicators.
Conclusion
- “Even if the overall symptoms are manageable, prophylactic treatment of OHE in such patients still needs to be considered”
I could not understand this sentence. OHE treatment was not started until clinical manifestation of OHE appeared in this study.
Reply: Apologies that this was not adequately clear. Although OHE symptoms can be controlled with pharmacological treatment, multiple unplanned hospitalizations actually increase medical expenses and health insurance burden. Therefore, the authors still recommend that prophylactic treatment should be emphasized after TIPS for patients with unembolized L-PUV.
- OHEs in this study were controlled using conservative therapy alone without LPUV embolization.
Therefore, even In TIPS cases with LPUV, OHE control was relatively easy with hepatic function being maintained and mortality rates being comparable to TIPS cases without LPUV. Post-interventional care after TIPS cases with LPUV might be easier than TIPS case with other SPSSs.
Reply: Many thanks. Despite our knowledge of the risk factors and preventive measures for HE, it was still the most common complication after TIPS. However, despite the high incidence, routine medical management was adequate for most patients, with or without L-PUV. on the other hand, whereas, the higher frequency of OHE in the L-PUV group means more hospitalizations and inevitably an increase in medical costs and medical burden. This is where we need to be concerned.
Round 2
Reviewer 3 Report
Thank you for your efforts to revise the manuscript. The revision manuscript may be generally improved at introduction, materials and methods, and result sections but remains insufficiency of the discussion section.
Some grammatical corrections are required (ex, Line 83)
We leave it to the editor's final discretion. If not, please consider re-submitting your request to another reviewer.
Author Response
Response to Reviewers for jcm-2016938 “Large Paraumbilical Vein Shunts Increase the Risk of Overt Hepatic Encephalopathy after Transjugular Intrahepatic Portosystemic Shunt Placement” by Hao-Huan Tang et al.
(original reviewers’ comments in black, our reponse in red)
Reviewer #3
Thank you for your efforts to revise the manuscript. The revision manuscript may be generally improved at introduction, materials and methods, and result sections but remains insufficiency of the discussion section.
Some grammatical corrections are required (ex, Line 83)
Reply: Many thanks for highlighting this. The discussion section and the grammar in the text have been further revised.
